# Seventy-Two-Hour LRRK2 Kinase Activity Inhibition Increases Lysosomal GBA Expression in H4, a Human Neuroglioma Cell Line

**DOI:** 10.3390/ijms23136935

**Published:** 2022-06-22

**Authors:** Clara Ruz, José Luis Alcantud, Francisco Vives, Francisco Arrebola, John Hardy, Patrick A. Lewis, Claudia Manzoni, Raquel Duran

**Affiliations:** 1Department of Physiology, Faculty of Medicine, Universidad de Granada, 18016 Granada, Spain; clararuz@ugr.es (C.R.); fvives@ugr.es (F.V.); 2Institute of Neurosciences “Federico Olóriz”, Centro de Investigación Biomédica (CIBM), Universidad de Granada, 18016 Granada, Spain; jlalcantud@ugr.es (J.L.A.); fav@ugr.es (F.A.); 3Department of Histology, Faculty of Medicine, Universidad de Granada, 18016 Granada, Spain; 4Department of Neurodegenerative Disease, UCL Queen Square Institute of Neurology, London WC1N 3BG, UK; j.hardy@ucl.ac.uk (J.H.); plewis@rvc.ac.uk (P.A.L.); 5Department of Comparative Biomedical Science, Royal Veterinary College, Royal College Street, London NW1 0TU, UK; 6Department of Pharmacology, UCL School of Pharmacy, London WC1N 1AX, UK; c.manzoni@ucl.ac.uk

**Keywords:** LRRK2, lysosomal dysfunction, glucocerebrosidase, Parkinson’s disease, autophagy

## Abstract

Mutations in LRRK2 and GBA1 are key contributors to genetic risk of developing Parkinson’s disease (PD). To investigate how LRRK2 kinase activity interacts with GBA and contributes to lysosomal dysfunctions associated with the pathology of PD. The activity of the lysosomal enzyme β-Glucocerebrosidase (GCase) was assessed in a human neuroglioma cell model treated with two selective inhibitors of LRKK2 kinase activity (LRRK2-in-1 and MLi-2) and a GCase irreversible inhibitor, condutirol-beta-epoxide (CBE), under 24 and 72 h experimental conditions. We observed levels of GCase activity comparable to controls in response to 24 and 72 h treatments with LRRK2-in-1 and MLi-2. However, GBA protein levels increased upon 72 h treatment with LRRK2-in-1. Moreover, LC3-II protein levels were increased after both 24 and 72 h treatments with LRRK2-in-1, suggesting an activation of the autophagic pathway. These results highlight a possible regulation of lysosomal function through the LRRK2 kinase domain and suggest an interplay between LRRK2 kinase activity and GBA. Although further investigations are needed, the enhancement of GCase activity might restore the defective protein metabolism seen in PD.

## 1. Introduction

Parkinson’s disease (PD) is the most common neurodegenerative movement disorder of old age [1]. While the majority of PD cases are idiopathic, research over the past two decades has revealed that a substantial minority of people with Parkinson’s develop the disorder due to the presence of genetic risk factors or causative mutations in their genomes [2]. Coding variants in two genes, *GBA1* on chromosome 1 and *LRRK2* on chromosome 12, are the most commonly identified genetic risk factors for PD [3,4]. Notably, both *GBA1* and *LRRK2* have also been identified as risk loci for idiopathic Parkinson’s by genome-wide association studies, suggesting a continuum of genetic risk between familial and idiopathic forms of disease [5]. The enzymatic activities of both GBA and LRRK2 are closely linked to their role in the etiology of Parkinson’s, with loss of glucocerebrosidase (GCase) activity due to mutations in *GBA1* and a likely gain of kinase activity due to mutations in *LRRK2* [6,7]. Intriguingly, recent research has suggested that there may be a reciprocal relationship between the enzymatic activities of LRRK2 and GBA, supporting a biological link between two of the most prominent risk factors for PD. The precise nature of this relationship is unclear. Cellular studies suggest that inhibition or loss of LRRK2 function increases GCase activity, and that mutations in *LRRK2*—thought to increase LRRK2 activity—result in decreased GCase activity [8], and that inhibiting LRRK2 boosts GCase activity in *GBA1* mutation cells [9]. Conversely, a study of dried blood spots from *LRRK2* mutation carriers suggested that carriers of the G2019S mutation harbored increased GCase activity [10]. This, in turn, contrasts with a further post-mortem analysis demonstrating decreased GBA protein levels in the brains of individuals with the G2019S mutation [11].

To investigate the relationship between LRRK2 and GBA, in this study, specific inhibitors of LRRK2 kinase activity and GCase activity were applied to a human neuroglioma cell line, assessing the consequence of inhibiting LRRK2 on GCase activity.

## 2. Results

To examine the impact of LRRK2 inhibition on GCase activity, H4 neuroglioma cells were exposed to 24 h and 72 h treatments with LRRK2in1 and MLi-2, two inhibitors of LRRK2 kinase activity [12,13]. Twenty-four-hour inhibition of LRRK2 kinase activity did not result in any significant alteration in GCase activity (Figure 1A). Following extended treatments with LRRK2in1 and MLi2 over 72 h, GCase activity was also comparable to DMSO-treated cells (Figure 1B). As expected, condutirol-beta-epoxide (CBE) treated cells showed reduced GCase activity in both experimental conditions. Moreover, GCase activity between 24 h versus 72 h treatments was compared (Appendix A). DMSO and LRRK2in1 treated cells showed a significant increase in GCase activity at 72 h in comparison to 24 h. To assess possible changes in protein expression levels after treatments with the inhibitors, immunoblot analysis of GBA was performed alongside the lysosomal marker LAMP1 and the marker for macroautophagy LC3 (Figure 1C,D). When 24 h treatment with LRRK2in1 was applied, only LC3-II displayed significantly increased protein levels compared to DMSO-treated cells (Figure 1I), while 72 h treatment resulted in significantly increased protein levels for both GBA and LC3-II when compared to DMSO-treated cells (Figure 1F,J) and LAMP1 when compared to CBE-treated cells (Figure 1H). However, the treatment with MLi-2 did not show significant changes in any of the three proteins for both experimental conditions (Figure 1E–J).

We also examined the phosphorylation status of Rab10 as a direct readout for LRRK2 kinase activity. As expected, treatments with LRRK2in1 and MLi2 decreased the phosphorylation of Rab10 (pRab10), confirming the efficacy of both inhibitors (Figure 2).

To recapitulate both results in enzymatic activity and in protein levels, we examined the magnitude of change in specific GCase activity, dividing the average GCase activity by the estimate of GBA protein quantification. Following 24 h treatment, control cells showed a GCase activity/GBA estimate ratio of 4.694 while cells treated with LRRK2in1 and MLi-2 inhibitors showed a GCase activity/GBA estimate ratio of 1.200 and 2.902, respectively. After 72 h treatment, control cells showed a GCase activity/GBA estimate ratio of 4.863, while cells treated with LRRK2in1 and MLi-2 inhibitors showed a GCase activity/GBA estimate ratio of 1.375 and 2.423, respectively (Figure 3).

To explore whether the observed decrease in specific GCase activity was due to an accumulation of immature GBA caused by defective protein trafficking/sorting through Golgi apparatus, cell lysates were subjected to Endo-H digestion. The results presented in Figure 4 showed no relevant changes in the Endo-H cleavage pattern, suggesting that most of the GBA protein is being correctly sorted to the mid Golgi and matured to lysosomal GBA.

## 3. Discussion

In this study, the impact of inhibiting LRRK2 kinase activity on GCase levels in a cell model for human neuroglioma was investigated. Twenty-four- and seventy-two-hour inhibition of LRRK2 kinase activity resulted in no significant increase in overall GCase activity when compared to DMSO-treated cells. However, a significant increase in GBA protein levels was observed following 72 h treatment with LRRK2in1. These data were combined by normalizing GCase activity levels to GBA protein levels, showing a striking reduced specific GCase activity after LRRK2 kinase inhibition. Concomitant with these changes, an increase in protein markers for macroautophagy activation (LC3-II) was observed. These data complement, and expand upon, a growing literature examining the link between LRRK2 and GBA1. At a genetic level, the population frequency of mutations in both genes, especially in specific populations such as the Ashkenazim and Imazighen, has resulted in a number of individuals harboring PD-associated mutations in *LRRK2* and *GBA1*, and there is evidence from the literature that supports an interaction at a clinical level between the impact of mutations in the two genes [14,15]. At a biochemical level, this has been examined in both human tissue and in a cellular context. Two studies reported analyses of GBA and GCase activity in human samples from carriers of mutations in *LRRK2*. Alcalay and co-workers examined GCase activity in dried blood spots from 36 carriers of the G2019S mutation, reporting that this was elevated compared to non-mutation carriers (13.69 μmol/L/h versus 11.93 μmol/l/h, *p* value 0.002) [10]. Zhao and colleagues investigated GBA protein levels in the brains of 17 individuals with the G2019S and I2020T mutations (*n* = 12 and 5, respectively), revealing a significantly reduced level of Tris-buffered-saline-soluble GBA protein in frontal cortex samples from *LRRK2* mutation carriers versus control [11]. A study of the relationship between LRRK2 and GCase activity in human neuronal cultures by Ysselstein and co-workers reported that cells carrying the G2019S mutation in LRRK2 had reduced GCase activity, and that inhibition of LRRK2 kinase activity increased GCase activity [8]. A further study by Sanyal and colleagues reported that inhibition of LRRK2 kinase activity in cells that harbor a mutation in *GBA1* boosts GCase activity [9].

The results from this current study suggest that there is a complex and multifaceted relationship between LRRK2 and GBA. First, there is a delay between inhibition of LRRK2 kinase activity and alterations in GBA, with changes only observed after 72 h of inhibition. This suggests that any links between LRRK2 and GBA are not immediately proximal to LRRK2 kinase activity, instead being a downstream event. Secondly, in the human neuroglioma cell model under examination, there were two outcomes of LRRK2 kinase inhibition upon GBA: an increase in GBA protein levels, and a decrease in the specific GCase activity of GBA. Intriguingly, the decrease in specific activity may mirror data from post-mortem studies of *LRRK2* mutation carriers, where the G2019S mutation is associated with decreased GBA protein levels [11] and increased GCase activity [10]. As the G2019S mutation causes an increase in LRRK2 kinase activity, the prediction from these studies would be that inhibition of LRRK2 kinase activity would result in an increase in GBA protein levels and a decrease in specific activity. In a recent work, Kedariti et al. showed that pharmacological inhibition of G2019S LRRK2 in HEK293T cells caused a significant reduction in GCase activity when it was normalized by GBA levels, being consistent with the results in our study [16].

The precise nature of the relationship between LRRK2 kinase activity and GBA GCase activity is not known. The requirement for 72 h inhibition of LRRK2 kinase activity to elicit a change in GBA biology suggests that there may be a number of steps separating LRRK2 and GBA within the cell. Moreover, it is interesting to clear up whether the selective inhibition of LRRK2-dependent phosphorylation on endo-lysosomal function can be influenced by the chemical structure of the inhibitors used. This could explain why treatment with MLi-2 does not exert the same effect as LRRK2in-1 on autophagy markers. Yao et al. observed differences in the efficacy and pharmacodynamics of two LRRK2 inhibitors, TTT-3002 and LRRK2in-1, upon evaluation of dopaminergic neurodegeneration caused by different *LRRK2* mutations using transgenic C. elegans models. Although both inhibitors were able to rescue the pathological phenotype manifested by the expression of mutants *LRRK2* G2019S and R1441C, results suggested that TTT-3002 showed higher affinity for LRRK2 proteins and higher potency of inhibition over kinase activity of LRRK2 [17]. Furthermore, Mercatelli et al. found that LRRK2in-1 and GSK2578215A, two structurally unrelated LRRK2 kinase activity inhibitors, differentially affected neurotransmitter release as well as pSer935 levels in synaptosomes from a mouse striatum and cerebral cortex [18]. These results would indicate that every inhibitor might have a different mode to interact with the LRRK2 kinase pocket. Furthermore, it remains unresolved whether the observed changes following LRRK2 kinase inhibitor treatment are entirely due to kinase activity inhibition or are also related to off-target effects. Due to the contribution of kinase proteins in the regulation of many subcellular functions, the design of highly selective inhibitors is essential to specifically block LRRK2 kinase activity. Recent works have identified hundreds of proteins susceptible to being phosphorylated by LRRK2 kinase in order to detect changes in their phosphorylation levels upon LRRK2 kinase activity inhibition [13,19,20]. Despite the widespread use of LRRK2in-1 and MLi-2 in exploring LRRK2 biology, their pharmacological impact on other molecular pathways should be further studied. Finally, given the reported role of LRRK2 in responding to lysosomal damage [21,22], one potential mechanism whereby inhibition of LRRK2 would result in alterations in GBA could be impairments in the basal turnover of damaged lysosomes, leading to accumulation of GBA protein and a paradoxical decrease in activity. Further investigations are required to clarify the mechanisms connecting LRRK2 and GBA. As the status of both LRRK2 and GBA as drug targets for PD is known, such investigations should be prioritized in the coming years.

## 4. Materials and Methods

### 4.1. Inhibitors

LRRK2in1 and MLi-2 were purchased from the Eurodiagnostico (HY-10875 100 mg and HY-100411 5 mg, respectively, Madrid, Spain). Condutirol-beta-epoxide (CBE, sc-201356) was purchased from Santa Cruz. All compounds used were dissolved in dimethyl sulfoxide (DMSO).

### 4.2. Antibodies

Antibodies used were as follows: rabbit anti-human LC3 antibody (NB100-2220, Novus Biologicals, Madrid, Spain), mouse anti-human LAMP-1 antibody [H4A3] (ab 25630, Abcam, Cambridge, United Kingdom), mouse anti-human β-Glucocerebrosidase antibody (GBA) (B-6) (sc-166407, Santa Cruz, Heidelberg, Germany), rabbit recombinant anti-rab10 (phospho T73) antibody (ab230261, Abcam), rabbit monoclonal anti-Rab10 (8127S, Cell Signaling, Danvers, MA, United States), mouse anti-human β-Actin antibody (A1978-200UL, Sigma, Merck Life Science, Madrid, Spain), anti-rabbit antibody (A0545, Sigma, Merck Life Science, Madrid, Spain) and anti-mouse antibody (A3682, Sigma).

### 4.3. Cell Culture and Pharmacological Treatments

H4 human neuroglioma cells were grown in DMEM containing 10% Fetal Bovine Serum (FBS). H4 cells (ATCC number HTB-148) were seeded into six-well plates (2 mL for each well) once they reached 90% confluence (2 × 10^5^ cells/mL approximately). After 24 h from plating, cells were treated with LRRK2in1, MLi-2 (both selective LRRK2 inhibitors with unrelated structures), CBE (an irreversible β-GCase enzyme inhibitor) or vehicle (DMSO as control) under 24 or 72 h treatments. For 72 h treatment, the medium was replaced with LRRK2in1, MLi-2 and CBE, and cells were incubated for 72 h. For short-term treatment, the medium was replaced as above, and cells were incubated for 3 days. Information about concentrations of every inhibitor is reported in Table 1. LRRK2in1 and CBE concentrations used for both experimental conditions were previously described and tested in several studies, finding no signs of toxicity for cells [23,24,25]. MLi-2 was used at a concentration of 600 nM based on previous works [13,26,27,28]. Cell viability was analyzed by Trypan blue assay. Appendix A shows representative images of MLi-2 or DMSO treated cells for 24 and 72 h. No changes in structural morphology and cell density were observed.

After treatment, cells were washed in Dulbecco’s phosphate buffered saline (DPBS), collected in RIPA lysis buffer: 150 Mm NaCl, 1.0% Triton X-100, 0.5% sodium deoxycholate, 0.1% SDS, 50 Mm Tris Ph 7.5, protease inhibitors (cOmplete, protease inhibitor cocktail, Roche) and phosphatase inhibitors (Halt phosphatase inhibitor cocktail, Pierce) and incubated on ice for 30 min. After centrifugation at 17,000× *g*, 4 °C for 20 min, the Triton-soluble supernatant was stored at −20 °C until analysis. Both experimental conditions (24 and 72 h treatments) were repeated to obtain independent replicates.

### 4.4. β-Glucocerebrosidase Enzymatic Assay

GCase activity was determined using the artificial substrate 4-Methylumbelliferyl (4-MU) β-d-glucopyranoside (M3633, Sigma Merck Life Science, Madrid, Spain) as previously described by Dijk et al. [29]. Briefly, the collected samples were incubated with the substrate in citrate phosphate buffer pH: 4.5 (0.1 M citric acid/0.2 M solution dibasic sodium phosphate) with 0.2% sodium taurodeoxycholate hydrate (W3026000, Sigma Aldrich) in a 96-well plate at 37 °C for 90 min. After incubation, the reaction was stopped with glycine-NaOH buffer 0.2 M pH: 10.4 (0.2 M solution Glycine/0.2 M NaOH). Fluorescence was measured using an Infinite M200Pro TECAN plate reader (Tecan Group Ltd., Männedorf, Switzerland) with 360 nm excitation wavelength and 446 nm emission wavelength. A standard curve of free 4-MU was used as reference to calculate the enzyme activity. Results were normalized to protein content.

### 4.5. Endo-H Treatment

An amount of 10 μg of proteins from cell lysates was subjected to one hour of incubation at 37 °C with Endoglycosidase-H (Endo-H) according to the manufacturer’s instructions (Endo H, P0702S, New England BioLabs, Hitchin, UK).

### 4.6. SDS-PAGE and Western Blot Analysis

Protein concentration of cell lysates was measured by BCA protein assay (Pierce BCA Protein Assay Kit, 23225 Thermo-Fisher Scientifics, Waltham, MA, USA). An amount of 10 μg of proteins was loaded in each lane of 4–12% Bis-Tris NuPAGE gels (Invitrogen, Thermo Fisher Scientific, Waltham, MA, USA) and separated according to the manufacturer’s instructions. After electrophoresis, proteins were transferred to PVDF membranes (IPVH00010, Immobilon -P (PVDF), Merck Millipore, Merck Life Science, Madrid, Spain) for 2 h. Membranes were blocked in 5%milk for 1 h and then incubated in the appropriate primary antibodies (diluted in Superblock solution, Thermo-Fisher Scientifics, Waltham, MA, USA) overnight. Bands were visualized after incubation with the respective horseradish peroxidase-linked secondary antibodies using Enhanced Chemiluminescence (ECL) in the digital imaging system Kodak Image Station 4000MM PRO. Then, bands were quantified using the ImageJ software (v. 1.52, NIH, https://imagej.nih.gov/ij/index.html accessed on 21 May 2022).

### 4.7. Statistical Analyses

All statistical analyses were performed with GraphPad Prism (GraphPad Software, San Diego, CA, USA).

## Figures and Tables

**Figure 1 ijms-23-06935-f001:**
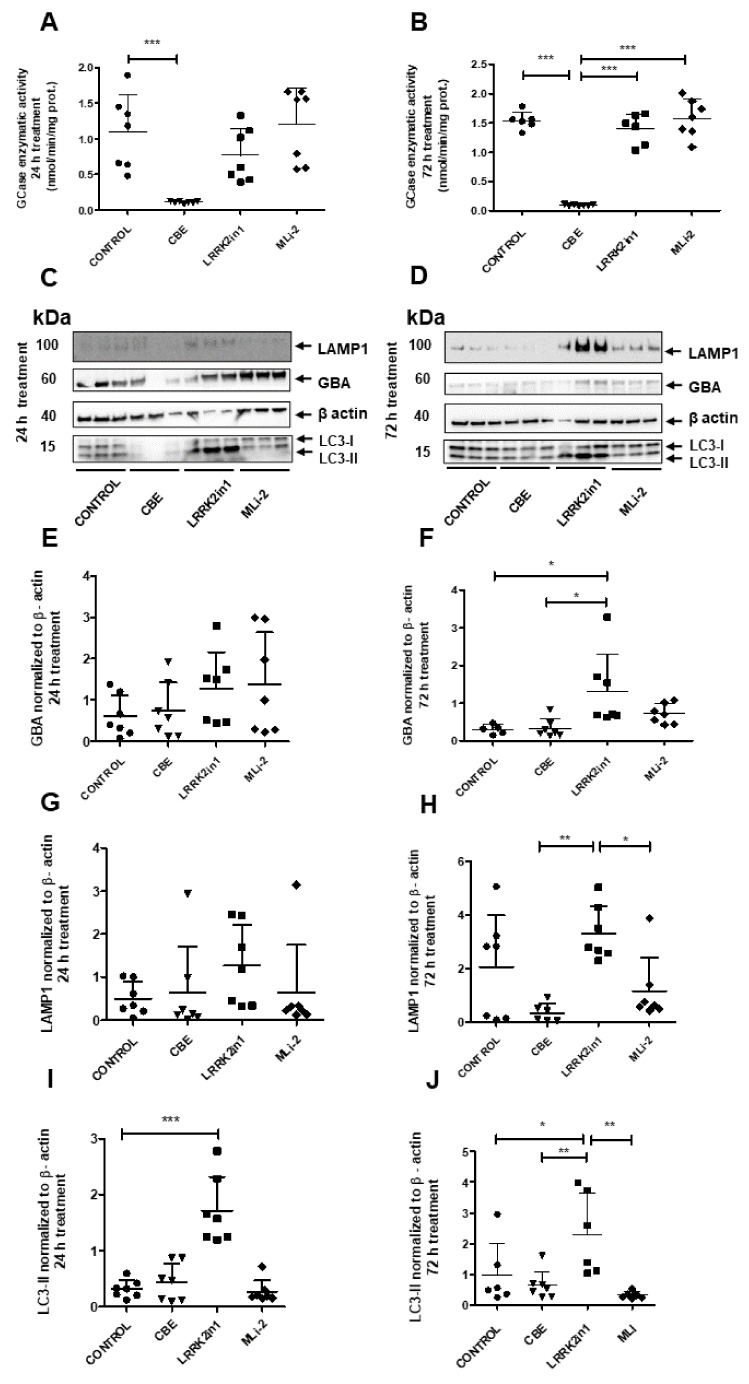
(**A**) Twenty-four-hour inhibition with both LRRK2-in1 and MLi2 produced no significant increase in GCase enzymatic activity compared to control cells. Quantifications were based on seven independent experiments. (**B**) Inhibition of LRRK2 kinase activity did not show significant changes in GCase activity in H4 neuroglioma cells after exposure to 72 h treatment when compared with control cells. Quantifications were based on seven independent experiments. (**C**,**D**) Immunoblot analysis of GBA, LAMP1 and LC3-II proteins from cell lysates 24 and 72 h treatments. Molecular weight markers are indicated in kilodaltons. (**E**–**J**) Quantification of GBA, LAMP1 and LC3-II protein levels in cell lysates after treatment normalized to loading control β-actin. Twenty-four-hour inhibition of the kinase activity of LRRK2 significantly increased the LC3-II protein levels when LRRK2in1 was applied (**I**). However, after 72 h inhibition, both LC3-II and GBA protein levels were significantly increased (**F**,**J**). Data of each experimental condition were grouped and normalized to perform analysis of variance followed by Tukey and Dunnett post hoc tests (values represent mean and SD, *** *p* < 0.001, ** *p* < 0.01, * *p* < 0.05).

**Figure 2 ijms-23-06935-f002:**
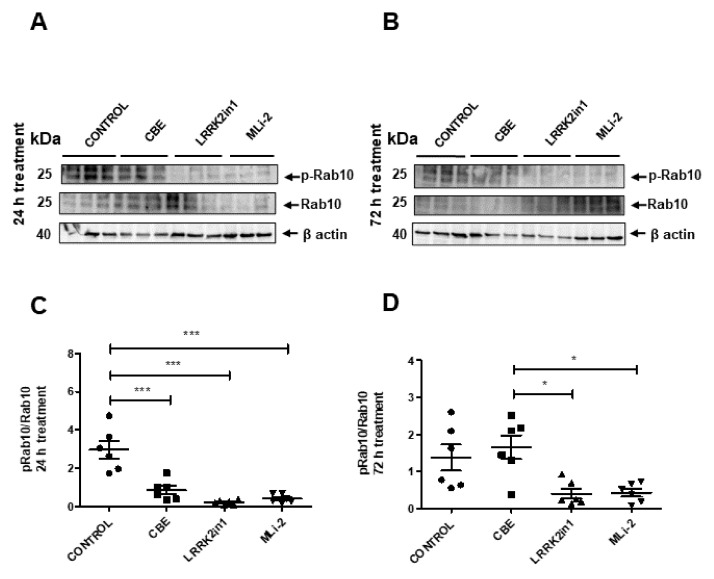
(**A**,**B**) Representative Western blot of pRab10 and total Rab10 in cell lysates treated with LRRK2in1, MLi2 and CBE. (**B**) Images were quantified by normalization to total Rab10 protein levels from six independent experiments. (**C**,**D**) Data demonstrated a significant reduction in the pRab10/Rab10 ratio in cell lysates treated with both LRRK2 inhibitors respect to control cells after 24 h treatment (Values represent mean and SD, *** *p* < 0.001, * *p* < 0.05).

**Figure 3 ijms-23-06935-f003:**
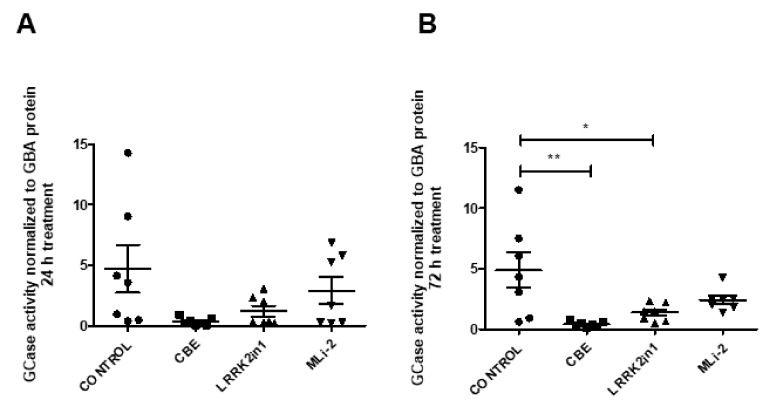
GCase activity/GBA estimate ratio of cell lysates treated with LRRK2in1, MLi2 and CBE ((**A**) 24 h treatment; (**B**) 72 h treatment). Specific GCase activity is significantly reduced in cell lysates after 72 h treatment with LRRK2in1 compared to DMSO-treated cells (Values represent mean and SD, ** *p* < 0.01, * *p* < 0.05.

**Figure 4 ijms-23-06935-f004:**
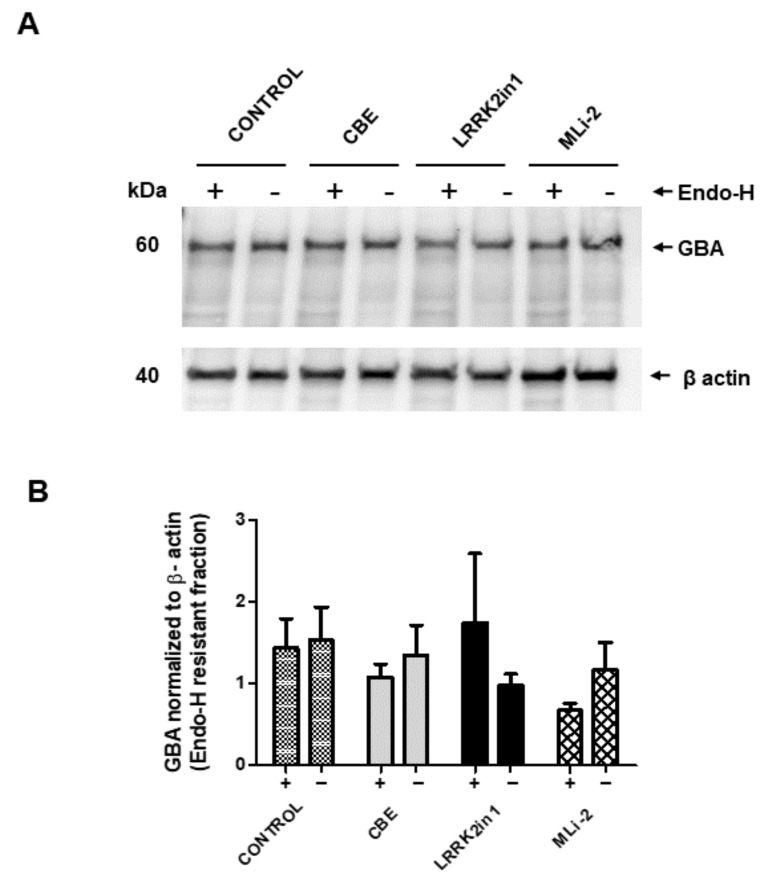
Endo-H resistance of GBA protein in cell lysates. (**A**) Cell lysates containing the same amount of protein were subjected to endo-H digestion and Western blot analysis with anti-GBA antibody. (**B**) Blots were scanned, and the intensity of each band was measured. Bars (+) represent Endo-H resistant fraction of GBA protein normalized to loading control β-actin from three independent experiments. Data show no significant differences in GBA protein levels between any of the treatments applied.

**Table 1 ijms-23-06935-t001:** Inhibitor concentrations for the 24 h and 72 h treatments.

72 h Treatment		
Inhibitor	Concentration	Pharmacological Effect
LRRK2in1	5 µM	LRRK2 kinase domain inhibitor
MLi-2	600 nM	LRRK2 kinase domain inhibitor
CBE	75 nM	β-GCase enzyme inhibitor
DMSO	0.15% *v*/*v*	Organic solvent
**24 h treatment**		
**Inhibitor**	**Concentration**	**Pharmacological effect**
LRRK2in1	5 µM	LRRK2 kinase domain inhibitor
MLi-2	600 nM	LRRK2 kinase domain inhibitor
CBE	75 nM	β-GCase enzyme inhibitor
DMSO	0.15% *v*/*v*	Organic solvent

## Data Availability

The data from GCase activity presented in this study are available on request from the corresponding author.

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
