# Peer review of "Seventy-Two-Hour LRRK2 Kinase Activity Inhibition Increases Lysosomal GBA Expression in H4, a Human Neuroglioma Cell Line"

_ijms, 2022, doi:10.3390/ijms23136935_

Round 1
Reviewer 1 Report
The paper investigates the effect of two LRRK2 inhibitors in the attempt to elucidate the biological interaction between LRRK2 and GBA1 through a series of biochemical and protein assays within a neuroglioma cell line.
The LRRK2 inhibitors seem to both inhibit LRRK2 activity towards its substrate Rab as shown by Figure 2. Nonetheless, the inhibitors exhibit very distinct effects on GCase activity and macroautophagy stimulation (LAMP1 & LC3-II). This presents confounding data and a limitation to the study as conclusions drawn may be related to off target effects of the inhibitor rather than LRRK2 inhibition.
Major Considerations
Line 67, 81, 125: Given the presented data, one cannot say that GCase activity increased with LRRK2in1 and MLi2 when compared to CBE-treated cells but rather was not significantly different than the control (DMSO-treated) cells. More evidently, CBE treatment is inhibiting GCase activity as expected rather than LRRK2 increasing GCase activity.
Bright’ study et al. [23] does not seem to justify the 600nM MLi-2 concentration used. Cell line viability data over the treatment and dose-response curves would be valuable here.
Manuscript does attempt to explain why the two inhibitors are presenting such distinct data but should delve into more detail and also mention the possibility of off-target effects.
Minor Recommendations
Title revision is suggested. The use of “Long-term” is discouraged as it is an ambiguous term that generally refers to weeks in culture not 72hrs. Also, it would be more accurate to specify “in H4, a human neuroglioma cell line” rather than just “in human neuroglioma cells”.
Long-term and short-term are ambiguous terms and should be changed to 72-hours and 24-hours throughout the manuscript.
Comment on the mutational status of LRRK2 and GBA in the H4 cell line and why the H4 cell line was selected.
Figure 1A&B – “enzimatic” activity should be “enzymatic”
Line 125: “in a cell model for human astrocytes” should be changed to “in a cell model for neuroglioma”. The model is not a normal human astrocyte cell model.
Line 204: (2 x 105 cells) needs to be (2 x 10^5 cells) or (2 x 105 cells)
Reviewer 2 Report
Ruz et al described LRRK2 inhibitor could increase lysosomal β-glucocerebrosidase activity in human neuroglioma cells. In my opinion, using “Long-term” for 72 hours can be easily misunderstood or misleading. Looking at the actually data between 24 and 72 hours, the control values variation may be contributed a lot to the weakly statistical differences. Statistical comparison between two controls as well as “24” vs “72” hours may be informative.
Round 2
Reviewer 1 Report
Authors have efficiently tackled most of the concerns reviewed and the additional discussion point regarding the inhibitors is appreciated.
An issue regarding the MLi-2 concentration still persists. The Volpicelli-Daley et al reference uses 1.1nM while Kluss et al uses MLi-2 administered to animals which would differ in concentration compared to cell models. Hence I would add the following reference: https://doi.org/10.1124/jpet.115.227587
The reference recommended also coincidentally assesses off-target effects of MLI-2 which may be useful contributions to the discussion.
